# Hyperpolarized ^13^C Magnetic Resonance Spectroscopic Imaging of Pyruvate Metabolism in Murine Breast Cancer Models of Different Metastatic Potential

**DOI:** 10.3390/metabo11050274

**Published:** 2021-04-27

**Authors:** Erin B. Macdonald, Paul Begovatz, Gregory P. Barton, Sarah Erickson-Bhatt, David R. Inman, Benjamin L. Cox, Kevin W. Eliceiri, Roberta M. Strigel, Suzanne M. Ponik, Sean B. Fain

**Affiliations:** 1Department of Medical Physics, University of Wisconsin-Madison, 1111 Highland Ave., Madison, WI 53705, USA; erin.beth.macdonald@gmail.com (E.B.M.); begovatz@wisc.edu (P.B.); gregory.barton@utsouthwestern.edu (G.P.B.); bcox1@wisc.edu (B.L.C.); eliceiri@wisc.edu (K.W.E.); rstrigel@uwhealth.org (R.M.S.); 2Morgridge Institute for Research, 330 N. Orchard St., Madison, WI 53715, USA; sarah.e.bhatt@gmail.com; 3Laboratory for Optical and Computational Instrumentation, University of Wisconsin-Madison, Madison, WI 53706, USA; 4Department of Cell and Regenerative Biology, University of Wisconsin-Madison, 1111 Highland Ave., Madison, WI 53705, USA; drinman@facstaff.wisc.edu (D.R.I.); ponik@wisc.edu (S.M.P.); 5Department of Biomedical Engineering, University of Wisconsin-Madison, 1415 Engineering Dr., Madison, WI 53706, USA; 6Carbone Cancer Center, University of Wisconsin-Madison, 600 Highland Ave., Madison, WI 53705, USA; 7Department of Radiology, University of Wisconsin-Madison, 600 Highland Ave., Madison, WI 53792, USA

**Keywords:** breast cancer, metastatic potential, metabolism, MRSI, hyperpolarized, carbon-13, repeatability

## Abstract

This study uses dynamic hyperpolarized [1-^13^C]pyruvate magnetic resonance spectroscopic imaging (MRSI) to estimate differences in glycolytic metabolism between highly metastatic (4T1, *n* = 7) and metastatically dormant (4T07, *n* = 7) murine breast cancer models. The apparent conversion rate of pyruvate-to-lactate (k_PL_) and lactate-to-pyruvate area-under-the-curve ratio (AUC_L/P_) were estimated from the metabolite images and compared with biochemical metabolic measures and immunohistochemistry (IHC). A non-significant trend of increasing k_PL_ (*p* = 0.17) and AUC_L/P_ (*p* = 0.11) from 4T07 to 4T1 tumors was observed. No significant differences in tumor IHC lactate dehydrogenase-A (LDHA), monocarboxylate transporter-1 (MCT1), cluster of differentiation 31 (CD31), and hypoxia inducible factor-α (HIF-1α), tumor lactate-dehydrogenase (LDH) activity, or blood lactate or glucose levels were found between the two tumor lines. However, AUC_L/P_ was significantly correlated with tumor LDH activity (*ρ*_spearman_ = 0.621, *p* = 0.027) and blood glucose levels (*ρ*_spearman_ = −0.474, *p* = 0.042). k_PL_ displayed a similar, non-significant trend for LDH activity (*ρ*_spearman_ = 0.480, *p* = 0.114) and blood glucose levels (*ρ*_spearman_ = −0.414, *p* = 0.088). Neither k_PL_ nor AUC_L/P_ were significantly correlated with blood lactate levels or tumor LDHA or MCT1. The significant positive correlation between AUC_L/P_ and tumor LDH activity indicates the potential of AUC_L/P_ as a biomarker of glycolytic metabolism in breast cancer models. However, the lack of a significant difference between in vivo tumor metabolism for the two models suggest similar pyruvate-to-lactate conversion despite differing metastatic potential.

## 1. Introduction

Breast cancer is the most common cancer diagnosis in women worldwide. This disease is responsible for the second most cancer-related deaths for women in developed countries and the most cancer-related deaths for women in developing countries [1]. Treatment of metastatic disease is still challenging in-part because tumor cells can remain dormant at distal sites for years to decades before emerging into overt metastatic outgrowth [2]. There is a critical need to identify and distinguish tumors that are highly metastatic from those that are metastatically dormant.

One hallmark of cancer is dysregulated energy metabolism [3]. As early as 1924, Otto Warburg observed that tumors and highly proliferative tissues exhibited upregulated glycolysis with increased pyruvate-to-lactate conversion compared with healthy normal tissues, independent of local oxygen availability [4]. While committing pyruvate to oxidative phosphorylation constitutes a more efficient means of energy metabolism, in cancer the lactate-dehydrogenase (LDH)-mediated conversion of pyruvate to lactate is hypothesized to enable production of glycolytic intermediates useful for building the macromolecules required for cell proliferation. Higher glycolytic rates may indicate more highly proliferative and aggressive tumors. Indeed, tumor lactate levels have been significantly positively correlated with the development of distant metastasis despite there being different lactate levels in tumors of the same grade and stage [5]. These results suggest that tumor lactate production may be an important biomarker for breast cancer aggressiveness and propensity to metastasize.

A promising technique for interrogating real-time in vivo metabolism is hyperpolarized ^13^C magnetic resonance spectroscopic imaging (MRSI). With this technique, ^13^C-labeled molecules of biological interest are exogenously hyperpolarized using dynamic nuclear polarization (DNP) to increase their MR signal from levels undetectable in a practical imaging timeframe to over 10,000-fold times higher [6]. The hyperpolarized ^13^C-labeled substrate is intravenously injected for imaging tumor metabolism in vivo, enabling the hyperpolarized ^13^C label to be observed as it is converted to downstream metabolites, each of which resonates at its own characteristic chemical shift frequency uniquely detected by the MR scanner [6].

Due to its important role in energy metabolism and amenability to hyperpolarization, [1-^13^C]pyruvate is the most commonly used substrate in hyperpolarized ^13^C magnetic resonance spectroscopy (MRS) and MRSI. Some of the earliest applications of hyperpolarized ^13^C MRS/MRSI exploited hyperpolarized [1-^13^C]pyruvate to monitor its LDH-mediated conversion to [1-^13^C]lactate in cancers, where glycolytic rates were expected to be high [7,8]. Hyperpolarized ^13^C MRS/MRSI has specific applications in breast cancer but with sometimes unexpected results. For example, Xu et al. explored differences in the apparent conversion rate of pyruvate-to-lactate (k_PL_) in mouse xenograft models of breast cancer with different levels of aggressiveness and found that more indolent tumors exhibited higher k_PL_ values, contrary to the conventional model of upregulated aerobic glycolysis predicted by the Warburg hypothesis [9,10]. In contrast, Ward et al. demonstrated the sensitivity of hyperpolarized pyruvate-to-lactate conversion to LDH modulation by inhibiting phosphoinositide 3-kinases (PI3K), a key enzyme implicated in cancer development and involved in regulating LDH concentrations [11]. An MYC-driven breast cancer tumor model was used to demonstrate decreased pyruvate-to-lactate conversion with reduced MYC-driven tumor progression and increased pyruvate-to-lactate conversion with cancer recurrence [12]. Despite the mechanistic complexity of breast cancers, the promise of hyperpolarized [1-^13^C]pyruvate MRS/MRSI has even fostered its initial translation into patients with invasive breast cancer [13] where the lactate/pyruvate ratio was positively and strongly correlated to monocarboxylate transporter 1 (MCT1) RNA expression suggesting cell membrane MCT1, rather than LDH activity, is rate-limiting in invasive breast cancers.

Dormant tumor cells, in other words, tumors that seed cells to distant sites but fail to form metastatic nodules, have been shown to disseminate at early tumor stages due to acquiring a highly motile, slowly proliferating phenotype [14]. While the mechanisms by which tumor cells acquire a dormant phenotype is not yet fully understood, markers of dormancy can be induced by hypoxia and dormant cells are characterized by reduced expression of the glucose transporter, Glut-1, and a shift in signaling from extracellular signal-related kinase (ERK) to p-38. However, in vitro studies using models of dormant tumor cells with the Seahorse flux analyzer method suggest the utilization of glucose through glycolysis is not significantly altered compared to metastatically aggressive cell lines [15]. In total, these characteristics suggest the dormant subpopulation of tumor cells might be identified by a unique metabolic signature stemming from tumor microenvironmental conditions in vivo, including possibly altered glycolysis due to local hypoxia [5]. However, the utility of hyperpolarized [1-^13^C]pyruvate MRSI as a prognostic marker to differentiate between breast cancers that are highly metastatic and those that are metastatic but dormant has not been directly explored.

In this work, we investigate the use of hyperpolarized [1-^13^C]pyruvate MRSI to differentiate between murine breast tumor xenografts of highly metastatic and metastatically dormant cells. Specifically, in vivo experiments were performed in highly metastatic 4T1 murine breast tumors and metastatically dormant 4T07 murine breast tumors that seed cells to the lungs and liver but do not form metastatic nodules. Dynamic hyperpolarized ^13^C MRSI data were used to estimate two measures of pyruvate to lactate conversion in these tumor models: (1) the apparent pyruvate-to-lactate conversion rate (k_PL_) and (2) the area-under-the-curve ratio of lactate-to-pyruvate (AUC_L/P_), a model-free metric known to be proportional to k_PL_ [16]. In exploratory studies, both k_PL_ and AUC_L/P_ were compared to independent measures of tumor LDH concentrations, immunohistochemistry (IHC), and serum lactate and glucose levels to study possible mechanisms mediating pyruvate-to-lactate conversion in these cell lines. Finally, a subset of mice underwent repeated testing 48 h later to gain insight into the variability of k_PL_ and AUC_L/P_ measures of pyruvate-to-lactate metabolism.

## 2. Results

### 2.1. Tumor Size

Estimated volumes and measured masses for both 4T07 and 4T1 tumor models at the last imaging time point are summarized in Table 1. The mean tumor volumes and masses were in good agreement between the metastatic dormant 4T07 and highly metastatic 4T1 tumor models. Variability in the 4T07 tumor dimensions was higher due to more variable growth rates compared to more rapidly proliferating 4T1 tumors.

### 2.2. Immunohistochemistry and Biochemical Analysis

Analysis of tumor tissue using immunohistochemistry revealed no differences in the presence of LDHA (*p* = 0.52), or the MCT1 transporter (*p* = 0.43) between metastatic 4T1 and metastatic-dormant 4T07 tumors (Figure 1). Furthermore, the presence of hypoxia inducible factor (HIF)-1α was investigated in order to identify changes in tumor lactate production which could be driven by hypoxia within the tumor microenvironment, rather than the anaerobic glycolysis; however, HIF-1α expression was also found to be equal between the tumor types (*p* = 0.52). Lastly, the area of CD31 immunohistochemistry was quantified to evaluate any differences in the presence of endothelial cells as an indicator of tumor vascularization. We found no change in CD31 area per tumor (*p* = 0.18), suggesting that vascularization and thus the delivery of hyperpolarized [1-^13^C]pyruvate was similar for all tumors. Mean IHC intensities are summarized in Figure 1 and Table 2 and representative images of each marker are shown in Appendix A.

No significant differences in tumor LDH activity, or blood lactate or glucose levels, were found between the two tumor lines (Table 2). Blood glucose levels did not significantly differ in mice bearing metastatic dormant 4T07 tumors compared to mice with more aggressive metastatic 4T1 tumors (*p* = 0.14).

### 2.3. Metabolic Imaging

Representative hyperpolarized [1-^13^C]pyruvate and [1-^13^C]lactate AUC images for each tumor model are displayed in Figure 2 along with their corresponding metabolite time courses and kinetic modeling fits. Boxplots of the resulting distribution of k_PL_ and AUC_L/P_ estimates for each tumor model are given in Figure 3. Mean metabolic imaging measures are summarized in Table 2, above. No significant differences in either metabolic imaging metric were found between the metastatic dormant 4T07 and metastatic 4T1 primary tumors. However, there is a trend of increasing k_PL_ (*p* = 0.17) and AUC_L/P_ (*p* = 0.11) for metastatic 4T1 tumors compared to metastatic dormant 4T07 tumors.

### 2.4. Correlations between Metabolic Measures

All Spearman correlation test results are summarized in Table 3. AUC_L/P_ was moderately correlated to k_PL_ (*ρ*_spearman_ = 0.67, *p* = 0.002) as expected (Figure A2). The k_PL_ measure was moderately repeatable with a coefficient of repeatability (CR k_PL_ = 0.04 s^−1^) and negligible bias (Figure A3 and Table A1). AUC_L/P_ was somewhat less repeatable (CR AUC = 1.2), with a larger coefficient of variation than k_PL_ (COV: AUC = 0.29 vs. k_PL_ = 0.23), but also showed negligible bias and no significant difference between time points.

Importantly, AUC_L/P_ was significantly correlated (*ρ*_spearman_ = 0.62, *p* = 0.027) with measured tumor LDH activity (Figure 4). k_PL_ displayed a similar trend of increasing value with increased tumor LDH activity, but the results did not reach significance (*ρ*_spearman_ = 0.48, *p* = 0.114).

The correlation plots comparing imaging measures of metabolism with blood lactate and glucose levels are displayed in Figure 5. Neither k_PL_ nor AUC_L/P_ had a significant correlation with blood lactate levels. However, AUC_L/P_ was significantly negatively correlated with blood glucose levels (*ρ*_spearman_ = −0.47, *p* = 0.042) and k_PL_ displayed a similar trend of decreasing value with increased blood glucose (*ρ*_spearman_ = −0.41, *p* = 0.088).

## 3. Discussion

Most breast cancer deaths are the result of metastatic disease rather than the primary tumor with many metastatic breast cancer patients succumbing to lactic acidosis [17], suggesting altered metabolism may play an important role in patient outcome. Indeed, upregulated glycolysis with increased pyruvate-to-lactate has been widely observed in tumors [4,18] including breast cancer [19]. Furthermore, tumor lactate levels have been significantly positively correlated with incidence of metastasis [5]. We used dynamic hyperpolarized ^13^C MRSI to investigate in vivo differences in pyruvate-to-lactate conversion (i.e., k_PL_ and AUC_L/P_) for metastatic dormant 4T07 and highly metastatic 4T1 tumors. Differences in k_PL_ and AUC_L/P_ did not reach statistical significance, suggesting similar pyruvate utilization in glycolysis for in vivo tumors, similar to in vitro findings using Seahorse flux analysis for the 4T1 and 4T07 cell lines [15]. Interestingly, there was a significant positive correlation between AUC_L/P_ and tumor LDH activity, and an observed trend towards higher AUC_L/P_ measures in the more metastatic 4T1 tumor model, but no significant differences were found in tumor LDH activity between the 4T07 and 4T1 tumors. This suggests that metastatic tumor potential is mediated by other metabolic processes such as increased glutamine metabolism [15]. Additionally, the trend of increased AUC_L/P_ and increased k_PL_ in more metastatic tumors may be influenced by other rate limiting steps in the conversion of pyruvate to lactate, such as hyperpolarized [1-^13^C]pyruvate delivery to the tumor (i.e., perfusion) [20], hyperpolarized [1-^13^C]pyruvate transport into the cell [19], the concentration of coenzyme nicotinamide adenine dinucleotide (NADH) [9], and the size of endogenous metabolite pools [21]. Follow-up histology and immunohistochemistry staining for microvascularity (CD31), MCT1, HIF-1α, and LDHA did not differ with metastatic potential either. Furthermore, tumor LDH activity and LDHA protein expression were not significantly correlated. While this result may seem unexpected, changes in LDH activity do not necessarily depend on the expression of LDHA. LDHA activity has been shown to be regulated by Src- or Her2-mediated phosphorylation of Y10 on LDHA [22]. Future work will address the possibility that LDHA activity in 4T1 or 4T07 tumors is regulated by phosphorylation.

Another in vivo study found that higher perfusion-corrected uptake of a fluorescent glucose derivative was observed in the metastatic 4T1 tumors compared to metastatic dormant 4T07 tumors for the same vascular oxygenation, indicating an increased shift towards glucose uptake via GLUT1 and presumably aerobic glycolysis in the highly metastatic 4T1 tumors [23]. However, this inference is not supported by our study, possibly because the pyruvate substrate follows a different path into the cell via monocarboxylate transporters (especially MCT1) in the cell membrane [24]. Indeed, in our study MCT1 neither differed between the tumor models or correlated to AUC_L/P_ or k_PL_, suggesting tumor LDH activity is a stronger driver of increased lactate production from the pyruvate substrate than MCT1 expression in these tumor models. This differs from recent findings in invasive human breast cancer that found stronger associations between pyruvate to lactate conversion and MCT1 expression than with LDH activity [19]. In this work, AUC_L/P_ was found to be significantly positively correlated with tumor LDH activity while k_PL_ displayed a similar, although non-significant, positive relationship. The stronger correlation between AUC_L/P_ and tumor LDH activity is likely attributable, at least in part, to its calculation from higher signal-to-noise ratio (SNR) data, with hyperpolarized ^13^C metabolite signals integrated over time as opposed to the time-resolved data points used for fitting k_PL_. Additionally, the goodness of fit of the kinetic model used to estimate k_PL_ depends on the choice of initial time frame with which to begin fitting, making it more susceptible to inaccuracies from noisy data than AUC_L/P_. Furthermore, pyruvate inflow affects the shape of the metabolite time courses and can bias metabolic measures. This work employed a two-way exchange model with an additional Heaviside step function term to model pyruvate inflow and help account for its influence on the fitted rate constants [25]. However, the Heaviside step function is an imperfect approximation of true pyruvate inflow and is unlikely to fully capture the true dynamics of pyruvate infusion. This lends additional appeal to AUC_L/P_, which is independent of both the pyruvate inflow function and the rates of pyruvate interconversion with metabolites other than lactate. Therefore, given that both k_PL_ and AUC_L/P_ are related to the same physiologic process, this work suggests AUC_L/P_ is a more sensitive biomarker of tumor LDH activity than k_PL_.

While the present work indicates potential for k_PL_ and AUC_L/P_ from hyperpolarized ^13^C MRSI of whole tumors, both tumor models used in this study are characterized by malignant primary tumors that seed metastatic nodules to distant sites. Therefore, differences in glycolysis may be too small for us to detect in a small study such as this, suggesting assessment of complementary metabolic pathways might show more pronounced differences as a complement to glycolytic rates. Both AUC_L/P_ and k_PL_ measures of pyruvate -to -lactate conversion are almost certainly influenced by more than just LDH activity. Concentration of coenzyme NADH [9], transport of hyperpolarized [1-^13^C]pyruvate via perfusion [20], and the size of endogenous metabolite pools [21] may all play a role in the observed metabolite dynamics.

Although upregulated glycolysis is a common feature associated with malignancy, recent works have suggested that mitochondrial oxidative phosphorylation may still be the dominant source of energy production in cancerous cells [18]. Scarcity of pyruvate substrate to fuel the tricarboxylic acid (TCA) cycle has been shown to provoke cancerous cells to use alternative sources, such as glutamine [26,27]. Furthermore, previous work by Xu et al. demonstrated that spatial patterns of increased mitochondrial redox state were predictive of tumor metastatic potential in breast cancer models, indicating mitochondrial metabolism may be more predictive of metastasis than cytosolic glycolysis alone [28]. Therefore, a more complete picture of differences in tumor metabolism between primary breast cancers of different metastatic potential may be gained by investigating the relative rates of glycolytic metabolism and oxidative phosphorylation. While the hyperpolarized ^13^C_1_ label on pyruvate is not transferrable to molecules in the TCA cycle, [2-^13^C]pyruvate and [1,2-^13^C]pyruvate have both been hyperpolarized to investigate glycolytic and mitochondrial metabolism simultaneously [29,30]. [5-^13^C]glutamine has also been successfully hyperpolarized for application in evaluating cancer metabolism [31] and may be a promising substrate for investigation in metastasis. Therefore, in addition to providing a potential biomarker for LDH-mediated conversion of pyruvate-to-lactate, hyperpolarized ^13^C MRSI offers the tools for a more comprehensive analysis of energy metabolism in breast cancer.

Finally, an unexpected trend observed in the present work was the tendency for blood glucose concentrations to be lower in the metastatic 4T1 tumor-bearing mice than the metastatic dormant 4T07 mouse models despite both sets of mice being consistently fasted for the same amount of time prior to imaging. While the direction of the association with blood glucose is at odds with ample literature demonstrating a link between blood glucose levels and increased risk of malignancy [32,33], the discrepancy may be reconciled by the fact that the tumor cells were injected into bilateral fat pads these mice as xenografts rather than occurring spontaneously. It is possible that, instead, the more aggressive and metastatic 4T1 tumors exert a higher energy demand on the systemic physiology than the metastatic dormant 4T07 tumors resulting in a hypoglycemic effect [34,35]. The energy demand from the rapidly growing 4T1 tumors may be further exacerbated by the burden of a contralateral tumor, although additional studies would be necessary to elucidate this trend. The significant negative correlation between AUC_L/P_ and blood glucose, and similarly negative trend between k_PL_ and blood glucose, suggest that increased systemic availability of glucose leads to decreased conversion of lactate to pyruvate in these 4T07 and 4T1 tumors. It is therefore possible that glucose acts as a competitive substrate for energy metabolism, reducing uptake of hyperpolarized [1-^13^C]pyruvate and, thereby, decreasing measures of pyruvate-to-lactate conversion. For example, previous work by Serrao et al. demonstrated lower variability in the measured hyperpolarized [1-^13^C]lactate/[1-^13^C]pyruvate ratio and k_PL_ in fasted animals where blood glucose levels are also expected to be more repeatable, indicating the influence of blood glucose on observed hyperpolarized [1-^13^C]pyruvate metabolism [36].

## 4. Materials and Methods

### 4.1. Cell Culture

The 4T1 and 4T07 murine carcinoma cells used to generate orthotopic mammary tumors were purchased from American Type Culture Collection (ATCC). Cells were maintained in culture at 5% CO_2_ with RPMI 1640 plus 10% FCS. Prior to orthotopic injection the cells were lifted, washed 3 times in sterile PBS, counted, and resuspended in sterile PBS at a final concentration of 1 × 10^7^ per mL for 4T1s and 2 × 10^7^ for 4T07s.

### 4.2. Animal Model and Handling

All animal experiments complied with Institutional Animal Care and Use Committee guidelines and requirements. A cohort of 14 female BALB/c mice (mass = 19–31 g) were imaged. Mice were divided evenly for bilateral injection with 50 µL of PBS containing either 5 × 10^5^ 4T1 cells or 1 × 10^6^ 4T07 murine breast cancer cells per inguinal mammary fat pad. These sample sizes were selected to approximately match those of Frees et al. who found a statistically significant difference in the delivery-corrected uptake of a fluorescent glucose derivative between these same orthotopic tumor models [23]. The 7 mice injected with 4T07 cells received 1 million cells per injection site except the first study which received ½ million cells per injection site. In all MRI experiments, 4T1 and 4T07 tumors were allowed to grow to a diameter of 0.5–1.0 cm. All mice were allowed unrestricted access to water but fasted for at least 4.5 h (mean ± standard deviation (SD) fasting times of 6.4 ± 0.6 h versus 6.5 ± 0.8 h for 4T07 and 4T1 mice, respectively) prior to anesthetization to achieve more reproducible blood-glucose levels at the time of metabolic imaging [36].

Just before imaging, mice were anesthetized with 3% isoflurane in oxygen (1 L/min) and maintained at 1–2.5% isoflurane throughout imaging. A cannula was placed in the tail vein for intravenous injection of hyperpolarized [1-^13^C]pyruvate during imaging. Internal body temperature and respiratory rate were continuously monitored using an intra-rectal fiber optic probe and respiratory pad, respectively. A warm air blower was placed approximately 5 cm from the tip of the mouse’s tail to heat the animal to 37 ± 1 °C during imaging. In a subset of 3 mice from each tumor line, a second hyperpolarized ^13^C MRSI study was performed 48 h after the initial experiment to assess repeatability of imaging measures of metabolism. The study design matrix is given in Table 4.

### 4.3. Biochemical and Immunohistochemical Analysis

Following each imaging study, and approximately 30 min after the hyperpolarized [1-^13^C]pyruvate injection (i.e., the approximate time to complete the study), ~30 μL of blood was drawn from the retro-orbital plexus while mice were still under anesthesia. Blood glucose and lactate levels were then measured using an Accu-Check Guide glucose meter (Roche, Basel, Switzerland) and a Lactate Plus lactate meter (Nova Biomedical, Waltham, MA, USA), respectively.

After the final imaging study, and following blood collection, mice were euthanized with CO_2_ and the tumors were harvested. Tumor length, width, and mass were recorded prior to any further tissue processing. Tumor volume was calculated from the length and width measurements [37]. Each tumor was divided for end-point analysis. For all mice except the first mouse bearing 4T07 tumors, one half of each tumor was snap frozen in liquid nitrogen and stored at −80 °C for LDH activity analysis. The remaining tumor halves for all mice were fixed in formalin for 48 h and then switched to 70% ethanol for histopathology.

Tumor LDH activity was measured only for the targeted imaging tumor in each mouse. Furthermore, 16.5–28.0 mg of snap frozen tumor tissue in 825–1400 μL of cold LDH assay buffer was homogenized on ice using an electric mixer. Samples were centrifuged at 10,000× *g* for 15 min at 4 °C. Moreover, 1 μL of serum was then assayed for LDH activity (Sigma-Aldrich MAK066, St. Louis, MO, USA) at a dilution of 1:10. Samples were compared to a standard curve generated from 2.5–12.5 nmol of reduced nicotinamide adenine dinucleotide (NADH) standards run at 450 nm and 37 °C.

Formalin fixed paraffin embedded tissue was used for immunofluorescence analysis of LDHA (ThermoFisher, Waltham, MA, USA, 1:500 dilution), HIF-1α (Invitrogen, Berlin, Germany, 1:200 dilution), MCT1 (ThermoFisher, Waltham, MA, USA, 1:400 dilution) and CD31 (Abcam, Cambridge, MA, USA, 1:100 dilution). Briefly, tumor sections were deparaffinized and rehydrated prior to antigen retrieval with citrate buffer [38]. Tissue was blocked for 1 h in 10% BSA (GeminiBio, West Sacramento, CA, USA) prior to incubation with primary antibody at room temperature. Tissue was then washed 3 times and incubated with the appropriate secondary antibody (Jackson Immuno Research, West Grove, PA USA, 1:500 dilution), counterstained with DAPI (Thermofisher, Waltham, MA, USA), and mounted with ProLong Gold (ThermoFisher, Waltham, MA, USA).

The tumor sections were imaged using a 20× Nikon air objective on a Nikon inverted Ti-300 Epifluorescent Microscope. A minimum of 5 images per individual tissue section were collected by random sampling of the tumor section using SlideBook 6.0 acquisition software and further analyzed with ImageJ. For all antibodies except CD31, the mean intensity was calculated for each image with ImageJ. Outliers identified by ROUT (GraphPad, Prism v9.1.0) were removed prior to averaging the intensity from all fields of view per tumor to determine differences between 4T1 and 4T07 tumors. In the case of CD31, a threshold was applied to create a mask of CD31 area per field of view. The area per field of view (FOV) was then averaged for each tumor to identify differences in CD31 area in 4T1 vs. 4T07 tumors.

The following exclusion criteria were applied. Blood samples were processed incorrectly for biochemical analysis of serum lactate and glucose levels in the first 4T07 tumor-bearing mouse, leaving 19/20 total samples for analysis. Additionally, snap frozen tumor tissue was also collected incorrectly for biochemical analysis of tumor LDH in the first 4T07 tumor-bearing mouse, leaving 13/14 tumors for analysis; tumors were harvested and compared to the last imaging time point only.

### 4.4. Hyperpolarization

Dynamic nuclear polarization (Hypersense, Tubney Woods, Abingdon, Oxfordshire, UK) was performed using 30 μL aliquots of [1-^13^C]pyruvic acid (Cambridge Isotope Laboratories Inc., Tewksbury, MA, USA) doped with 15 mM OX063 trityl radical (Oxford Instruments, Concord, MA, USA). Samples were irradiated with microwaves of 94.075 GHz and 100 mW to solid-state polarizations of over 98%. The sample was then rapidly dissoluted in a 4-mL neutralizing solution of 128 mM NaOH, 140 mM Tris buffer, and 88 mg/L EDTA. A dose of 10 μL/g of hyperpolarized sample was injected into the tail vein over 12–15 s during dynamic image acquisition. Liquid-state polarization of the remaining sample was measured in a bench-top polarimeter (^13^C-MQC polarimeter, Oxford Instruments Molecular Biotools Ltd., Abingdon, Oxfordshire, UK), giving polarizations of 18 ± 3% (mean ± SD) at the time of injection. The final [1-^13^C]pyruvate concentration and pH were ~110 mM and 7.8 ± 0.2 (mean ± SD), respectively.

### 4.5. Image Acquisition

All imaging was performed on a 4.7 T small animal MRI (Agilent, Palo Alto, CA, USA) using a dual-tuned ^1^H/^13^C volume coil for ^1^H imaging and ^13^C excitation, and a ^13^C surface coil for signal reception (Doty Scientific, Columbia, SC, USA). The ^13^C surface coil was centered over one tumor to ensure relatively homogeneous signal reception in that tumor. The selected tumor was alternated between mice when possible, such that 8 right tumors and 6 left tumors were imaged across the 14 mice. Tumors were imaged when they measured 0.5–1.0 cm in diameter to minimize the chance of necrosis and ulceration.

Shimming was performed on the ^1^H channel and multi-echo, ^1^H spoiled-gradient echo (SPGR) data (TR/TE_1_/ΔTE = 34.0/4.2/0.4 ms, FOV = 48 × 48 mm^2^, matrix = 192 × 192, slice thickness = 2 mm, flip angle = 20°, echoes = 8) were collected in order to generate a B_0_ field map using an image-space IDEAL reconstruction [39] for spatially-resolved frequency corrections in the direct IDEAL ^13^C reconstruction [40]. Additionally, ^1^H, T_2_-weighted fast spin-echo images (FOV = 48 × 48 mm^2^, matrix = 192 × 192, slice thickness = 2 mm, TR/TE_eff_ = 3500/66 ms, echo-train length = 8) were acquired for anatomical reference.

For ^13^C MRSI, power and frequency calibrations were performed with a thermally polarized phantom of 9.4 M ^13^C-urea (Cambridge Isotope Laboratories Inc., Tewksbury, MA, USA) doped with 8.5 mM gadobenate dimeglumine (Bracco Diagnostics, Inc., Princeton, NJ, USA) placed in the imaging field of view (FOV). Approximately 20 s prior to injection of hyperpolarized [1-^13^C]pyruvate, dynamic, constant-density, k-t spiral acquisitions were started with the following parameters: TR/TE_1_ = 50–150/0.318 ms, NE = 6, flip angle = 10°, and receiver bandwidth = 250 kHz. To achieve 1–3 pixels per tumor diameter, the nominal in-plane resolution was 3 × 3 mm^2^ (i.e., nominal FOV = 48 × 48 mm^2^ and nominal matrix = 16 × 16) and the slice thickness was 5 mm. A FOV-oversampling factor of η = 7 was used, resulting in a prescribed FOV and matrix of 336 × 336 mm^2^ and 112 × 112, respectively. For all ^13^C spiral acquisitions, the maximum gradient slew rate was derated to maintain a 30 ms readout for high acquisition SNR efficiency [41] assuming a realistic T_2_* [42,43,44]. Slice-selective spectra (flip angle = 5°, receiver bandwidth = 5 kHz) were interleaved between ^13^C image acquisitions to guide selection of metabolite peak frequencies in the direct IDEAL reconstruction. In total, 32 time frames were acquired at ~5 s temporal resolution following each injection.

### 4.6. Image Reconstruction

Metabolite images of hyperpolarized [1-^13^C]pyruvate, [1-^13^C]lactate, [1-^13^C]alanine, [1-^13^C]pyruvate-hydrate, and thermally-polarized ^13^C-urea were reconstructed using a direct IDEAL reconstruction in MATLAB (R2015b, The MathWorks, Natick, MA, USA) interfacing with C++. Reconstructions incorporated the estimated ^1^H B_0_ field maps, divided by a factor of four to account for the difference in ^1^H and ^13^C gyromagnetic ratios [40]. Gradient trajectory imperfections were also corrected using a thin slice-based technique [45]. Metabolite peak frequencies used in the reconstruction were identified as follows.

The relative offset of the excitation center frequency from [1-^13^C]pyruvate is difficult to decouple from B_0_ inhomogeneities influencing the hyperpolarized ^13^C slice-selective spectra. To resolve these two sources of off-resonance, the image of ^13^C metabolism was compared with the ^1^H SPGR anatomical image using mutual information while varying the chemical shift of [1-^13^C]pyruvate and [1-^13^C]lactate at 4.7 T to be 614 ± 20 Hz using an exhaustive search. The frequency combination used in the final reconstruction of all 32 ^13^C imaging time frames was the one that maximized the mutual information between the anatomy and the [1-^13^C]pyruvate and [1-^13^C]lactate that fell within the specified 614 ± 20 Hz range. [1-^13^C]alanine, [1-^13^C]pyruvate-hydrate, and ^13^C-urea were assumed to be at their known chemical shifts relative to [1-^13^C]pyruvate at 4.7 T (i.e., 433 Hz, 272 Hz, and −366 Hz from the [1-^13^C]pyruvate frequency, respectively).

### 4.7. Image Analysis

All image analysis was performed in MATLAB (R2015b, The MathWorks, Natick, MA, USA). Dynamic images of metabolism were first baseline corrected by subtracting the mean signal from time frames prior to hyperpolarized [1-^13^C]pyruvate injection. Voxel-wise area-under-the-curve ratios of lactate-to-pyruvate (AUC_L/P_) were then calculated [16] and the mean AUC_L/P_ for each tumor was found using a manually segmented region of interest (ROI) from a single individual. These same ROIs were used to calculate the mean hyperpolarized [1-^13^C]pyruvate and [1-^13^C]lactate signals in each tumor for each time frame. The resulting metabolite time courses were also fit to the two-way exchange model with a Heaviside step function to incorporate pyruvate inflow [25] in order to estimate the apparent exchange rate of pyruvate-to-lactate (k_PL_). For the first 4T1 tumor-bearing mouse, ^13^C MRSI metabolite time courses lacked sufficient SNR for kinetic modeling to estimate k_PL_, resulting in k_PL_ being estimated for 19/20 total imaging experiments. Due to the insufficient metabolite SNR in the contralateral tumors for reliable kinetic modeling, and the variability in positioning of the contralateral tumors with respect to the ^13^C surface-receive coil, imaging measures of metabolism are only presented for the primary tumors. Additionally, [1-^13^C]alanine and [1-^13^C]pyruvate-hydrate tumor signals were at or near the level of the noise in the metabolite images and, therefore, were not analyzed in this work.

### 4.8. Statistical Analysis

All statistical tests were performed in R 3.5.0 [46] unless otherwise stated. Unpaired, two-tailed *t*-tests were used to assess differences in k_PL_, AUC_L/P_, tumor LDH activity, and blood lactate and glucose levels between metastatic 4T1 and metastatic dormant 4T07 tumor models. For IHC, statistical analysis was performed using GraphPad Prism V6 (GraphPad Software, San Diego, CA, USA). A Shapiro–Wilk test for normality was performed, followed by Mann–Whitney test for normal data, or an unpaired *t*-test when data failed the normality test. Only imaging and biochemical metabolic markers from the first imaging day were used when comparing the two tumor models whereas immunohistochemistry stains were only available from the last imaging time point for each mouse. All available data from 4T1 and 4T07 tumor models on all imaging days were combined for correlation analysis. A non-parametric Spearman correlation test was used to compare the imaging metrics, k_PL_ and AUC_L/P_, with the following biochemical and IHC markers: tumor LDH activity, blood lactate and glucose levels, and tissue expression of LDHA, MCT1, HIF-1α, and CD31. Additionally, since AUC_L/P_ is meant to be a model-free parameter proportional to k_PL_, the correlation between AUC_L/P_ and k_PL_ was also tested. Imaging metrics and biochemical and IHC markers were also tested for correlation with tumor volume. Lastly, a correlation test was performed between tumor LDH activity and tumor LDHA expression. Results were considered significant for α ≤ 0.05.

## 5. Conclusions

Consistent with in vitro studies, hyperpolarized [1-^13^C]pyruvate MRSI measures of in vivo glycolytic flux, namely, k_PL_ and AUC_L/P_, exhibited an increasing trend between metastatic dormant (4T07) and highly metastatic (4T1) murine breast cancer models, although differences were not statistically significant. Importantly, a significant positive correlation was found between AUC_L/P_ and tumor LDH activity, further supporting AUC_L/P_ as a biomarker of in vivo glycolytic pyruvate-to-lactate conversion rate. IHC showed that MCT1 and LDHA did not correlate with k_PL_ and AUC_L/P,_ suggesting [1-^13^C]pyruvate MRSI is mostly driven by LDH enzyme activity in this model.

## Figures and Tables

**Figure 1 metabolites-11-00274-f001:**
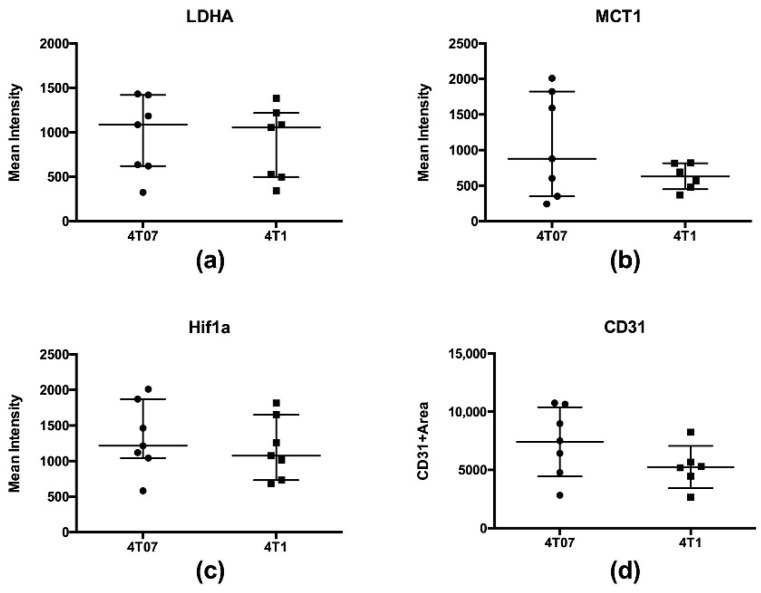
Comparison of immunohistochemistry results between 4T07 and 4T1 tumors: (**a**) LDHA; (**b**) MCT1 transporter; (**c**) HIF-1α; (**d**) and presence of endothelial cells via CD31. Error bars represent the median, 25th_,_ and 75th percentiles.

**Figure 2 metabolites-11-00274-f002:**
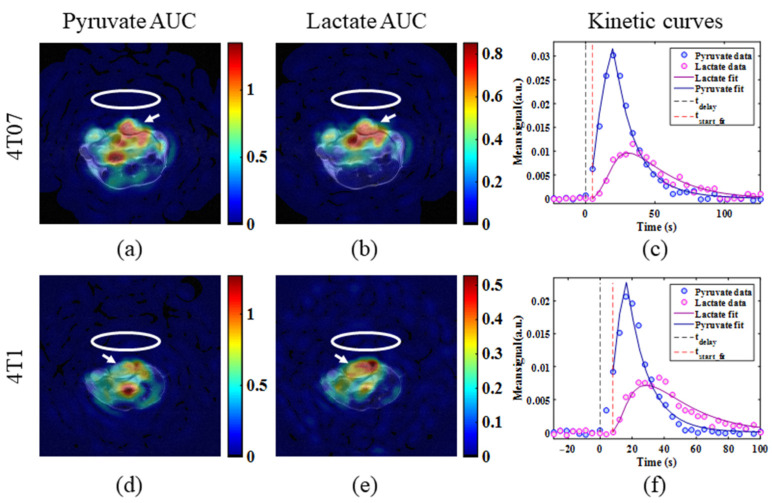
Representative hyperpolarized (**a**,**d**) [1-^13^C]pyruvate and (**b**,**e**) [1-^13^C]lactate AUC images (color) overlaid on T_2_-weighted anatomical reference images (grayscale) for mice bearing (**a**,**b**) metastatic dormant 4T07 tumors and (**d**,**e**) more aggressive, metastatic 4T1 tumors. The approximate ^13^C surface coil location and targeted tumor for imaging are indicated by white ovals and white arrows, respectively. The corresponding pyruvate and lactate metabolite time courses from the targeted imaging tumor are displayed for the (**c**) 4T07 and (**f**) 4T1 breast cancer models along with the fitted curves. The time of hyperpolarized [1-^13^C]pyruvate injection is indicated by t_delay_ and the first time point used for kinetic modeling is labeled with t_start_fit_.

**Figure 3 metabolites-11-00274-f003:**
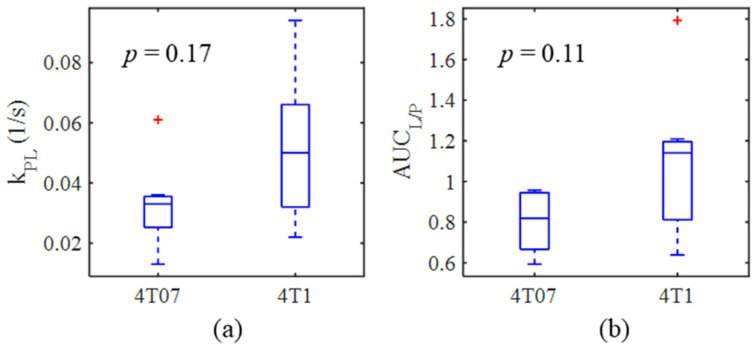
Boxplots comparing the distribution of (**a**) k_PL_ and (**b**) AUC_L/P_ between metastatic dormant 4T07 and highly metastatic 4T1 tumor models. Box plots represent the median with upper and lower limits representing the 25th and 75th percentiles, whiskers the minimum and maximum, and red crosses the outliers.

**Figure 4 metabolites-11-00274-f004:**
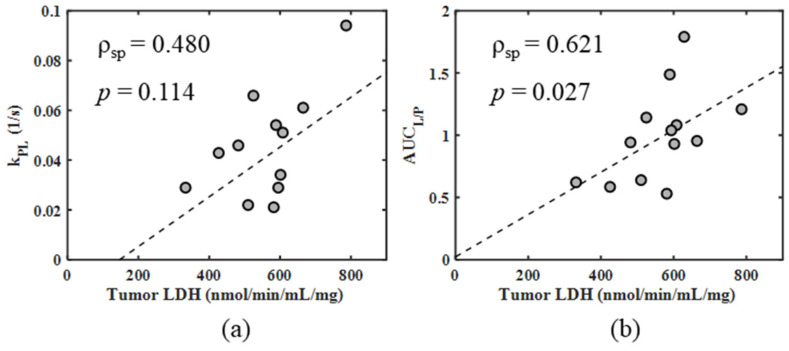
Spearman correlation plots comparing (**a**) k_PL_ and (**b**) AUC_L/P_ with tumor LDH activity. Spearman correlation coefficients (*ρ*_sp_) and *p*-values are listed on the plots.

**Figure 5 metabolites-11-00274-f005:**
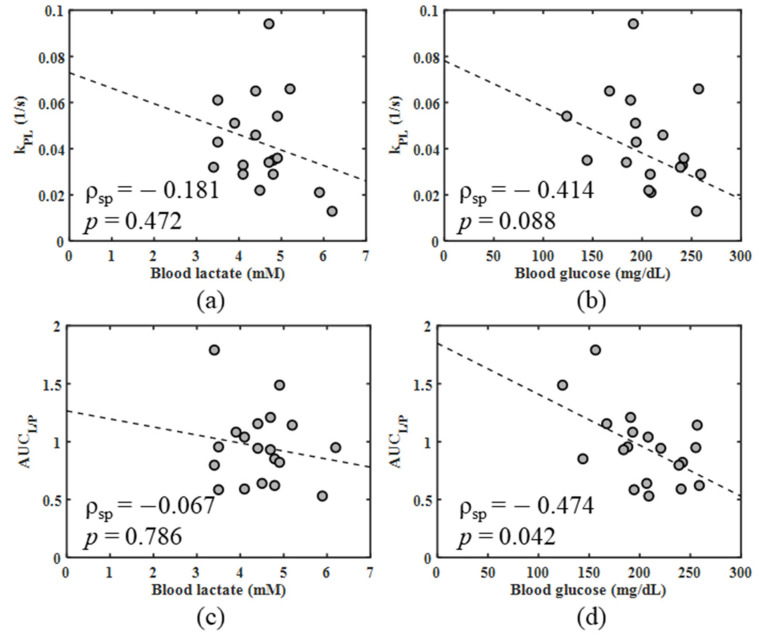
Spearman correlation plots comparing k_PL_ with blood (**a**) lactate and (**b**) glucose levels, and AUC_L/P_ with blood (**c**) lactate and (**d**) glucose levels. Spearman correlation coefficients (*ρ*_sp_) and *p*-values are listed on the plots.

**Table 1 metabolites-11-00274-t001:** Estimated tumor volumes and measured tumor masses for both metastatic dormant 4T07 and metastatic proliferative 4T1 tumor lines.

Tumor Model	*n*	Volume (mm^3^) Mean ± SD	Mass (mg) Mean ± SD	Days Post-Inoculation Mean ± SD
4T07	7	184 ± 178	163 ± 145	21 ± 8
4T1	7	195 ± 84	150 ± 58	12 ± 1

**Table 2 metabolites-11-00274-t002:** Statistical results comparing imaging, biochemical, and immunohistochemical metabolic measures between 4T07 and 4T1 tumor models.

Metabolic Measure	4T07		4T1		Unpaired *t*-Test
	Mean ± SD	# Samples	Mean ± SD	# Samples	
k_PL_ (s^−1^)	0.033 ± 0.015	7	0.052 ± 0.027	6	*p* = 0.17
AUC_L/P_	0.81 ± 0.15	7	1.08 ± 0.38	7	*p* = 0.11
Blood lactate (mM)	4.7 ± 0.9	6	4.3 ± 0.7	7	*p* = 0.45
Blood glucose (mg/dL)	228 ± 33	6	194 ± 43	7	*p* = 0.14
Tumor LDH (nmol/min/mL/mg)	535 ± 128	6	588 ± 102	7	*p* = 0.44
Tumor LDHA (mean intensity)	958 ± 433	7	872 ± 408	7	*p* = 0.52
Tumor MCT1 (mean intensity)	1071 ± 729	7	623 ± 184	6	*p* = 0.18
Tumor HIF-1α (mean intensity)	1329 ± 496	7	1176 ± 432	7	*p* = 0.52
Tumor CD31 (CD31 + area)	7422 ± 2973	7	5253 ± 1814	6	*p* = 0.43

Number of samples (# samples), area under the curve (AUC), lactate dehydrogenase-A (LDHA), monocarboxylate transporter-1 (MCT1), cluster of differentiation 31 (CD31), and hypoxia inducible factor-α (HIF-1α).

**Table 3 metabolites-11-00274-t003:** Spearman correlation results.

Comparison	Tumor Models Included	Time Points Included ^a^	*ρ* _spearman_	*p*-Value
k_PL_ vs. tumor LDH	pooled 4T1 and 4T07	final	0.480	0.114
k_PL_ vs. blood lactate	pooled 4T1 and 4T07	all	−0.181	0.472
k_PL_ vs. blood glucose	pooled 4T1 and 4T07	all	−0.414	0.088
k_PL_ vs. tumor LDHA	pooled 4T1 and 4T07	final	0.459	0.115
k_PL_ vs. tumor MCT1	pooled 4T1 and 4T07	final	−0.403	0.172
k_PL_ vs. tumor HIF-1α	pooled 4T1 and 4T07	final	−0.163	0.593
k_PL_ vs. tumor CD31	pooled 4T1 and 4T07	final	−0.233	0.443
k_PL_ vs. tumor volume	pooled 4T1 and 4T07	final	−0.201	0.511
AUC_L/P_ vs. tumor LDH	pooled 4T1 and 4T07	final	0.621	0.027
AUC_L/P_ vs. blood lactate	pooled 4T1 and 4T07	all	−0.067	0.786
AUC_L/P_ vs. blood glucose	pooled 4T1 and 4T07	all	−0.474	0.042
AUC_L/P_ vs. tumor LDHA	pooled 4T1 and 4T07	final	−0.057	0.844
AUC_L/P_ vs. tumor MCT1	pooled 4T1 and 4T07	final	−0.479	0.083
AUC_L/P_ vs. tumor HIF-1α	pooled 4T1 and 4T07	final	−0.275	0.342
AUC_L/P_ vs. tumor CD31	pooled 4T1 and 4T07	final	0.416	0.139
AUC_L/P_ vs. tumor CD31	pooled 4T1 and 4T07	final	−0.002	1.000
AUC_L/P_ vs. k_PL_	pooled 4T1 and 4T07	all	0.665	0.002
LDHA vs. tumor volume	pooled 4T1 and 4T07	final	−0.284	0.325
MCT1 vs. tumor volume	pooled 4T1 and 4T07	final	0.351	0.239
HIF-1α vs. tumor volume	pooled 4T1 and 4T07	final	−0.095	0.750
CD31 vs. tumor volume	pooled 4T1 and 4T07	final	−0.024	0.940
LDHA vs. LDH activity	pooled 4T1 and 4T07	final	0.066	0.835

^a^ “Time points included” refers to the imaging time point data used from each mouse to perform the correlation test. “final” refers to the final imaging experiment for each mouse, some of which had repeat experiments. “all” refers to all imaging experiments, including the repeat experiments.

**Table 4 metabolites-11-00274-t004:** Study design matrix.

Tumor Model	Total Mice Imaged	Subset of Mice with Repeat Study
4T07, dormant	7	3
4T1, metastatic	7	3
Total	14	6

## Data Availability

Data are available in a publicly accessible repository at: https://www.medphysics.wisc.edu/research/pulmonary-imaging/. Please contact the corresponding author for further details.

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
