# Peer review of "Hyperpolarized 13C Magnetic Resonance Spectroscopic Imaging of Pyruvate Metabolism in Murine Breast Cancer Models of Different Metastatic Potential"

_metabolites, 2021, doi:10.3390/metabo11050274_

Round 1

Reviewer 1 Report

This study from Macdonald et al. explores metabolic imaging of two breast cancer models, 4T1 and 4T07, in order to infer changes in metabolic flux with respect to aggressiveness. They observe a trend toward increased [1-13C]lactate in the 4T1 model as compared to the 4T07 though no measurement comparing the groups (of any measurement in the paper) is statistically significant. While I agree there is quite a lot of work put into this study, it is challenging for me to see what the relevant conclusion of this paper would be.

  1. For many of the non-standard measurements in this paper (IHC mean intensities for LDHA and MCT1) the SD is nearly 50% of the mean. Detecting any change would require a massive effect given the lack of precision here. I would suggest the authors consider other methods if they aim to infer anything about these proteins.
  2. A correlation is drawn between tumor LDH and AUC of lactate/pyruvate (AUClp). This seems to be at odds with the measurements of LDH activity on its own for the 2 tumor types. Additionally kpl and AUClp are the most significantly correlated (if I’m not mistaken the most significant correlation in the entire paper), yet kpl is not correlated to LDH activity? To better characterize what is going on, I would suggest the authors consider an in vivo tracing experiment with 13C pyruvate and a complimentary analysis.

Minor comments:

  1. The abstract states that the authors “…estimate differences in glycolytic metabolism between seven highly metastatic (4T1) and seven metastatically dormant (4T07) murine breast cancer models.” This is of course not true. There are only 2 models with n=7 mice in each group, each mouse does not represent a different metastatic model. This should be clarified.
  2. The authors acquire 6 echoes to reconstruct their images using a slightly modified IDEAL. Given that pyruvate is known to be metabolized to multiple products (alanine, bicarbonate etc) why did they not reconstruct with those known metabolites?
  3. The mice have bilateral tumors which are clearly visible in the images and appear to have adequate SNR in the AUC overlays. Why would the authors choose to not quantify the fluxes in those tumors? Ratios would be reasonably invariant to issues related to not being in the perfect center of the surface coil.
  4. The authors should consider the effect of 2 tumors and differential burden on the mice when comparing system wide measurements (e.g. blood glucose, lactate etc.). Also it is unclear as to how necrotic or hemorrhaged these tumors are at the time of imaging. The faster growing tumor may not be all viable tumor…

Author Response

Thank you for the considered and thorough assessment of our work. Please find attached responses to Reviewer 1. 

Reviewer 2 Report

Recent developments in hyperpolarization techniques allow dramatic increases in sensitivity of orders of magnitude that open new perspectives for metabolic imaging both in preclinical and in clinical application.

In this contribution, a well-constructed and well-written piece of work, the authors observed the d-DNP-produced hyperpolarized (HP) [1-13C]pyruvate metabolism to estimate differences in glycolytic metabolism between highly metastatic and metastatically dormant murine breast cancer models. The authors report the comparison between different metabolic measurements methods: conversion rate of pyruvate-to-lactate (kPL), lactate-to-pyruvate area-under-the-curve (AUC) ratio, ex-vivo biochemical metabolic measures and immunohistochemistry.

The results obtained using the HP [1-13C]pyruvate, in particular the AUCs, indicate the potential of this HP metabolite as a biomarker of glycolytic metabolism in breast cancer models, despite the difference in pyruvate-to-lactate conversion were similar in the two tumour models.

The work is undoubtedly interesting, carefully done and nicely presented.

In my opinion, the paper should certainly be published as is subject to minor corrections, which I hope the authors will find helpful.

  • Page 1, line 19-21: I think it would be appropriate to change this sentence removing both “seven”.

At first reading, it would seem that the authors compared seven types of metastatic tumour models (starting from 4T1 cells) and seven low-metastatic types starting from 4T07 cells.

If the authors want to specify in the abstract the number of animals used, they could simply add “(N = 7)” in the text.

  • Page 3, Table 1: The size of the tumours would be more useful for future readers if it were indicated what day after inoculation they refer to.

If the authors have this information, I suggest adding a graph with the growth curves of the two tumour models.

  • Page 6, Table 3: Authors should indicate the meaning of the second column of the Table. Reading the rest of the work it is understood that some mice were given pyruvate injection twice and some information was excluded as discussed on page 10 (line 359-365), but this information should be provided in a more usable way for future readers.

  • Page 9, line 293: In what are the cells resuspended for injection? PBS, RPMI, other?

  • Page 9, line 322: Which criterion was used to establish that the sampling of the blood occurred 30 min after the injection of HP pyruvate? Is this just the time it takes to finish the imaging study?

  • Page 14, Table A1: In my opinion, it would be more interesting to show the values ​​for the two tumour models separately, even if they do not show a significant difference.

Author Response

Thank you for the considered and thorough assessment of our work. Please find attached responses to Reviewer 2. 

Round 2

Reviewer 1 Report

The authors have provided responses to my previous questions. I would encourage them to tone down their findings in the discussion and discuss the limitations of their methods. More specifically that they can not rule out any of the other mechanisms given as they can't really measure them with sufficient accuracy. Additionally they did not modulate any of those parameters and therefore can't actually say which governs the changes in lactate they observe.